# IMPLICIT BIAS OF LARGE DEPTH NETWORKS: A NOTION OF RANK FOR NONLINEAR FUNCTIONS

**Arthur Jacot**
Courant Institute of Mathematical Sciences
New York University
New York, NY 10012, USA
`arthur.jacot@nyu.edu`

## ABSTRACT

We show that the representation cost of fully connected neural networks with homogeneous nonlinearities - which describes the implicit bias in function space of networks with $L_2$-regularization or with losses such as the cross-entropy - converges as the depth of the network goes to infinity to a notion of rank over nonlinear functions. We then inquire under which conditions the global minima of the loss recover the 'true' rank of the data: we show that for too large depths the global minimum will be approximately rank 1 (underestimating the rank); we then argue that there is a range of depths which grows with the number of datapoints where the true rank is recovered. Finally, we discuss the effect of the rank of a classifier on the topology of the resulting class boundaries and show that autoencoders with optimal nonlinear rank are naturally denoising.

## 1 INTRODUCTION

There has been a lot of recent interest in the so-called implicit bias of DNNs, which describes what functions are favored by a network when fitting the training data. Different network architectures (choice of nonlinearity, depth, width of the network, and more) and training procedures (initialization, optimization algorithm, loss) can lead to widely different biases.

In contrast to the so-called kernel regime where the implicit bias is described by the Neural Tangent Kernel (Jacot et al., 2018), there are several active regimes (also called rich or feature-learning regimes), whose implicit bias often feature a form sparsity that is absent from the kernel regime. Such active regimes have been observed for example in DNNs with small initialization (Chizat & Bach, 2018; Rotskoff & Vanden-Eijnden, 2018; Li et al., 2020; Jacot et al., 2022a), with $L_2$-regularization (Savarese et al., 2019; Ongie et al., 2020; Jacot et al., 2022b) or when trained on exponentially decaying losses (Gunasekar et al., 2018a;b; Soudry et al., 2018; Du et al., 2018; Ji & Telgarsky, 2018; Chizat & Bach, 2020; Ji & Telgarsky, 2020). In the latter two cases, the implicit bias is described by the representation cost:

$$R(f) = \min_{\mathbf{W}:f_{\mathbf{W}}=f} \|\mathbf{W}\|^2$$

where $f$ is a function that can be represented by the network and the minimization is over all parameters $\mathbf{W}$ that result in a network function $f_{\mathbf{W}}$ equal to $f$, the parameters $\mathbf{W}$ form a vector and $\|\mathbf{W}\|$ is the $L_2$-norm.

The representation cost can in some cases be explicitly computed for linear networks. For diagonal linear networks, the representation cost of a linear function $f(x) = w^T x$ equals the $L_p$ norm $R(f) = L \|w\|_p^p$ of the vector $v$ for $p = \frac{2}{L}$ (Gunasekar et al., 2018a; Moroshko et al., 2020) where $L$ is the depth of the network. For fully-connected linear networks, the representation cost of a linear function $f(x) = Ax$ equals the $L_p$-Schatten norm (the $L_p$ norm of the singular values) $R(f) = L \|A\|_p^p$ (Dai et al., 2021).

A common thread between these examples is a bias towards some notion of sparsity: sparsity of the entries of the vector $w$ in diagonal networks and sparsity of the singular values in fully connected

networks. Furthermore, this bias becomes stronger with depth and in the infinite depth limit $L \to \infty$ the rescaled representation cost $R(f)/L$ converges to the $L_0$ norm $\|w\|_0$ (the number of non-zero entries in $w$) in the first case and to the rank $\mathrm{Rank}(A)$ in the second.

For shallow ($L = 2$) nonlinear networks with a homogeneous activation, the representation cost also takes the form of a $L_1$ norm (Bach, 2017; Chizat & Bach, 2020; Ongie et al., 2020), leading to sparsity in the effective number of neurons in the hidden layer of the network.

However, the representation cost of deeper networks does not resemble any typical norm ($L_p$ or not), though it still leads to some form of sparsity (Jacot et al., 2022b). Despite the absence of explicit formula, we will show that the rescaled representation cost $R(f)/L$ converges to some notion of rank in nonlinear networks as $L \to \infty$, in analogy to infinite depth linear networks.

## CONTRIBUTIONS

We first introduce two notions of rank: the Jacobian rank $\mathrm{Rank}_J(f) = \max_x \mathrm{Rank}\,[Jf(x)]$ and the Bottleneck rank $\mathrm{Rank}_{BN}(f)$ which is the smallest integer $k$ such that $f$ can be factorized $f = h \circ g$ with inner dimension $k$. In general, $\mathrm{Rank}_J(f) \leq \mathrm{Rank}_{BN}(f)$, but for functions of the form $f = \psi \circ A \circ \phi$ (for a linear map $A$ and two bijections $\psi$ and $\phi$), we have $\mathrm{Rank}_J(f) = \mathrm{Rank}_{BN}(f) = \mathrm{Rank}A$. These two notions of rank satisfy the properties (1) $\mathrm{Rank}f \in \mathbb{Z}$; (2) $\mathrm{Rank}(f \circ g) \leq \min\{\mathrm{Rank}f, \mathrm{Rank}g\}$; (3) $\mathrm{Rank}(f + g) \leq \mathrm{Rank}f + \mathrm{Rank}g$; (4) $\mathrm{Rank}(x \mapsto Ax + b) = \mathrm{Rank}A$.

We then show that in the infinite depth limit $L \to \infty$ the rescaled representation cost of DNNs with a general homogeneous nonlinearity is sandwiched between the Jacobian and Bottleneck ranks:

$$ \mathrm{Rank}_J(f) \leq \lim_{L \to \infty} \frac{R(f)}{L} \leq \mathrm{Rank}_{BN}(f). $$

Furthermore $\lim_{L \to \infty} R(f)$ satisfies properties (2-4) above. We also conjecture that the limiting representation cost equals its upper bound $\mathrm{Rank}_{BN}(f)$.

We then study how this bias towards low-rank functions translates to finite but large depths. We first show that for large depths the rescaled norm of the parameters $\|\hat{\mathbf{W}}\|^2/L$ at any global minimum $\hat{\mathbf{W}}$ is upper bounded by $1 + C_N/L$ for a constant $C_N$ which depends on the training points. This implies that the resulting function has approximately rank 1 w.r.t. the Jacobian and Bottleneck ranks.

This is however problematic if we are trying to fit a 'true function' $f^*$ whose 'true rank' $k = \mathrm{Rank}_{BN}f^*$ is larger than 1. Thankfully we show that if $k > 1$ the constant $C_N$ explodes as $N \to \infty$, so that the above bound ($\|\hat{\mathbf{W}}\|^2/L \leq 1 + C_N/L$) is relevant only for very large depths when $N$ is large. We show another upper bound $\|\hat{\mathbf{W}}\|^2/L \leq k + C/L$ with a constant $C$ independent of $N$, suggesting the existence of a range of intermediate depths where the network recovers the true rank $k$.

Finally, we discuss how rank recovery affects the topology of decision boundaries in classification and leads autoencoders to naturally be denoising, which we confirm with numerical experiments.

## RELATED WORKS

The implicit bias of deep homogeneous networks has, to our knowledge, been much less studied than those of either linear networks or shallow nonlinear ones. (Ongie & Willett, 2022) study deep networks with only one nonlinear layer (all others being linear). Similarly (Le & Jegelka, 2022) show a low-rank alignment phenomenon in a network whose last layers are linear.

Closer to our setup is the analysis of the representation cost of deep homogeneous networks in (Jacot et al., 2022b), which gives two reformulations for the optimization in the definition of the representation cost, with some implications on the sparsity of the representations, though the infinite depth limit is not studied.

A very similar analysis of the sparsity effect of large depth on the global minima of $L_2$-regularized networks is given in (Timor et al., 2022), however, they only show how the optimal weight matrices are almost rank 1 (and only on average), while we show low-rank properties of the learned function, as well as the existence of a layer with almost rank 1 hidden representations.

## 2 PRELIMINARIES

In this section, we define fully-connected DNNs and their representation cost.

### FULLY CONNECTED DNNS

In this paper, we study fully connected DNNs with $L+1$ layers numbered from $0$ (input layer) to $L$ (output layer). Each layer $\ell \in \{0, \ldots, L\}$ has $n_\ell$ neurons, with $n_0 = d_{in}$ the input dimension and $n_L = d_{out}$ the output dimension. The pre-activations $\tilde{\alpha}_\ell(x) \in \mathbb{R}^{n_\ell}$ and activations $\alpha_\ell(x) \in \mathbb{R}^{n_\ell}$ of the layers of the network are defined inductively as

$$\alpha_0(x) = x$$
$$\tilde{\alpha}_\ell(x) = W_\ell \alpha_{\ell-1}(x) + b_\ell$$
$$\alpha_\ell(x) = \sigma\left(\tilde{\alpha}_\ell(x)\right),$$

for the $n_\ell \times n_{\ell-1}$ connection weight matrix $W_\ell$, the $n_\ell$ bias vector $b_\ell$ and the nonlinearity $\sigma : \mathbb{R} \to \mathbb{R}$ applied entrywise to the vector $\tilde{\alpha}_\ell(x)$. The parameters of the network are the collection of all connection weights matrices and bias vectors $\mathbf{W} = (W_1, b_1, \ldots, W_L, b_L)$.

We call the network function $f_\mathbf{W} : \mathbb{R}^{d_{in}} \to \mathbb{R}^{d_{out}}$ the function that maps an input $x$ to the pre-activations of the last layer $\tilde{\alpha}_L(x)$.

In this paper, we will focus on homogeneous nonlinearities $\sigma$, i.e. such that $\sigma(\lambda x) = \lambda\sigma(x)$ for any $\lambda \geq 0$ and $x \in \mathbb{R}$, such as the traditional ReLU $\sigma(x) = \max\{0, x\}$. In our theoretical analysis we will assume that the nonlinearity is of the form $\sigma_a(x) = \begin{cases} x & \text{if } x \geq 0 \\ ax & \text{otherwise} \end{cases}$ for some $\alpha \in (-1, 1)$, since for a general homogeneous nonlinearity $\sigma$ (which is not proportional to the identity function, the constant zero function or the absolute function), there are scalars $a \in (-1, 1)$, $b \in \mathbb{R}$ and $c \in \{+1, -1\}$ such that $\sigma(x) = c\sigma_a(bx)$; as a result, the global minima and representation cost are the same up to scaling.

*Remark* 1. By a simple generalization of the work of (Arora et al., 2018), the set of functions that can be represented by networks (with any finite widths and depth) with such nonlinearities is the set of piecewise linear functions with a finite number of linear regions. In contrast, the three types of homogeneous nonlinearities we rule out (the identity, the constant, or the absolute value) lead to different sets of functions: the linear functions, the constant functions, or the piecewise linear functions $f$ such that $\lim_{t \to \infty} \|f(tx) - f(-tx)\|$ is finite for all directions $x \in \mathbb{R}^{d_{in}}$ (or possibly a subset of this class of functions). While some of the results of this paper could probably be generalized to the third case up to a few details, we rule it out for the sake of simplicity.

*Remark* 2. All of our results will be for sufficiently wide networks, i.e. for all widths $\mathbf{n}$ such that $n_\ell \geq n_\ell^*$ for some minimal widths $n_\ell^*$. Moreover these results are $O(0)$ in the width, in the sense that above the threshold $n_\ell^*$ the constants do not depend on the widths $n_\ell$. When there are a finite number of datapoints $N$, it was shown by (Jacot et al., 2022b) that a width of $N(N+1)$ is always sufficient, that is we can always take $n_\ell^* = N(N+1)$ (though it is observed empirically that a much smaller width can be sufficient in some cases). When we are trying to fit a piecewise linear function over the whole input domain $\Omega$, the width required depends on the number of linear regions (He et al., 2018).

### REPRESENTATION COST

The representation cost $R(f; \Omega, \sigma, L)$ is the squared norm of the optimal weights $\mathbf{W}$ which represents the function $f_{|\Omega}$:

$$R(f; \Omega, \sigma, L) = \min_{\mathbf{W}: f_{\mathbf{W}|\Omega} = f_{|\Omega}} \|\mathbf{W}\|^2$$

where the minimum is taken over all weights $\mathbf{W}$ of a depth $L$ network (with some finite widths $\mathbf{n}$) such that $f_\mathbf{W}(x) = f(x)$ for all $x \in \Omega$. If no such weights exist, we define $R(f; \Omega, \sigma, L) = \infty$.

The representation cost describes the natural bias on the represented function $f_\mathbf{W}$ induced by adding $L_2$ regularization on the weights $\mathbf{W}$:

$$\min_\mathbf{W} C(f_\mathbf{W}) + \lambda \|\mathbf{W}\|^2 = \min_f C(f) + \lambda R(f; \Omega, \sigma, L)$$

for any cost $C$ (defined on functions $f : \Omega \mapsto \mathbb{R}^{d_{out}}$) and where the minimization on the right is over all functions $f$ that can be represented by a depth $L$ network with nonlinearity $\sigma$. Therefore, if we can give a simple description of the representation cost of a function $f$, we can better understand what type of functions $f$ are favored by a DNN with nonlinearity $\sigma$ and depth $L$.

*Remark* 3. Note that the representation cost does not only play a role in the presence of $L_2$-regularization, it also describes the implicit bias of networks trained on an exponentially decaying loss, such as the cross-entropy loss, as described in (Soudry et al., 2018; Gunasekar et al., 2018a; Chizat & Bach, 2020).

## 3  INFINITELY DEEP NETWORKS

In this section, we first give 4 properties that a notion of rank on piecewise linear functions should satisfy and introduce two notions of rank that satisfy these properties. We then show that the infinite-depth limit $L \to \infty$ of the rescaled representation cost $R(f; \Omega, \sigma_a, L)/L$ is sandwiched between the two notions of rank we introduced, and that this limit satisfies 3 of the 4 properties we introduced.

### RANK OF PIECEWISE LINEAR FUNCTIONS

There is no single natural definition of rank for nonlinear functions, but we will provide two of them in this section and compare them. We focus on notions of rank for piecewise linear functions with a finite number of linear regions since these are the function that can be represented by DNNs with homogeneous nonlinearities (this is a Corollary of Theorem 2.1 from (Arora et al., 2018), for more details, see Appendix E.1). We call such functions finite piecewise linear functions (FPLF).

Let us first state a set of properties that any notion of rank on FPLF should satisfy, inspired by properties of rank for linear functions:

1. The rank of a function is an integer $\mathrm{Rank}(f) \in \mathbb{N}$.
2. $\mathrm{Rank}(f \circ g) \leq \min\{\mathrm{Rank}f, \mathrm{Rank}g\}$.
3. $\mathrm{Rank}(f + g) \leq \mathrm{Rank}f + \mathrm{Rank}g$.
4. If $f$ is affine ($f(x) = Ax + b$) then $\mathrm{Rank}f = \mathrm{Rank}A$.

Taking $g = id$ or $f = id$ in (2) implies $\mathrm{Rank}(f) \leq \min\{d_{in}, d_{out}\}$. Properties (2) and (4) also imply that for any bijection $\phi$ on $\mathbb{R}^d$, $\mathrm{Rank}(\phi) = \mathrm{Rank}(\phi^{-1}) = d$.

Note that these properties do not uniquely define a notion of rank. Indeed we will now give two notions of rank which satisfy these properties but do not always match. However any such notion of rank must agree on a large family of functions: Property 2 implies that $\mathrm{Rank}$ is invariant under pre- and post-composition with bijections (see Appendix A), which implies that the rank of functions of the form $\psi \circ f \circ \phi$ for an affine function $f(x) = Ax + b$ and two (piecewise linear) bijections $\psi$ and $\phi$ always equals $\mathrm{Rank}A$.

The first notion of rank we consider is based on the rank of the Jacobian of the function:

**Definition 1.** The Jacobian rank of a FPLF $f$ is $\mathrm{Rank}_J(f; \Omega) = \max_x \mathrm{Rank}Jf(x)$, taking the max over points where $x$ is differentiable.

Note that since the jacobian is constant over the linear regions of the FPLF $f$, we only need to take the maximum over every linear region. As observed in (Feng et al., 2022), the Jacobian rank measures the intrinsic dimension of the output set $f(\Omega)$.

The second notion of rank is inspired by the fact that for linear functions $f$, the rank of $f$ equals the minimal dimension $k$ such that $f$ can be written as the composition of two linear function $f = g \circ h$ with inner dimension $k$. We define the bottleneck rank as:

**Definition 2.** The bottleneck rank $\mathrm{Rank}_{BN}(f; \Omega)$ is the smallest integer $k \in \mathbb{N}$ such that there is a factorization as the composition of two FPLFs $f_{|\Omega} = (g \circ h)_{|\Omega}$ with inner dimension $k$.

The following proposition relates these two notions of rank:

**Proposition 1.** *Both* $\mathrm{Rank}_J$ *and* $\mathrm{Rank}_{BN}$ *satisfy properties* $1 - 4$ *above. Furthermore:*

- *For any FPLF and any set $\Omega$, $\mathrm{Rank}_J(f;\Omega) \leq \mathrm{Rank}_{BN}(f;\Omega)$.*

- *There exists a FPLF $f : \mathbb{R}^2 \to \mathbb{R}^2$ and a domain $\Omega$ such that $\mathrm{Rank}_J(f;\Omega) = 1$ and $\mathrm{Rank}_{BN}(f;\Omega) = 2$.*

INFINITE-DEPTH REPRESENTATION COST

In the infinite-depth limit, the (rescaled) representation cost of DNNs $R_\infty(f;\Omega,\sigma_a) = \lim_{L\to\infty} \frac{R(f;\Omega,\sigma_a,L)}{L}$ converges to a value 'sandwiched' between the above two notions of rank:

**Theorem 1.** *For any bounded domain $\Omega$ and any FPLF $f$*

$$\mathrm{Rank}_J(f;\Omega) \leq R_\infty(f;\Omega,\sigma_\alpha) \leq \mathrm{Rank}_{BN}(f;\Omega).$$

*Furthermore the limiting representation cost $R_\infty(f;\Omega,\sigma_a)$ satisfies properties 2 to 4.*

*Proof.* The lower bound follows from taking $L \to \infty$ in Proposition 3 (see Section 4). The upper bound is constructive: a function $f = h \circ g$ can be represented as a network in three consecutive parts: a first part (of depth $L_g$) representing $g$, a final part (of depth $L_h$) representing $h$, and in the middle $L - L_g - L_h$ identity layers on a $k$-dimensional space. The contribution to the norm of the parameters of the middle part is $k(L - L_g - L_h)$ and it dominates as $L \to \infty$, since the contribution of the first and final parts are finite. $\qquad\square$

Note that $R_\infty(f;\Omega,\sigma_a)$ might satisfy property 1 as well, we were simply not able to prove it. Theorem 1 implies that for functions of the form $f = \psi \circ A \circ \phi$ for bijections $\psi$ and $\phi$, $R_\infty(f;\Omega,\sigma_a) = \mathrm{Rank}_J(f;\Omega) = \mathrm{Rank}_{BN}(f;\Omega) = \mathrm{Rank}A$.

*Remark* 4. Motivated by some aspects of the proofs and a general intuition (which is described in Section 4) we conjecture that $R_\infty(f;\Omega,\sigma_a) = \mathrm{Rank}_{BN}(f;\Omega)$. This would imply that the limiting representation cost does not depend on the choice of nonlinearity, as long as it is of the form $\sigma_a$ (which we already proved is the case for functions of the form $\psi \circ A \circ \phi$).

This result suggests that large-depth neural networks are biased towards function which have a low Jacobian rank and (if our above mentioned conjecture is true) low Bottleneck rank, much like linear networks are biased towards low-rank linear maps. It also suggests that the rescaled norm of the parameters $\|\mathbf{W}\|^2/L$ is an approximate upper bound on the Jacobian rank (and if our conjecture is true on the Bottleneck rank too) of the function $f_\mathbf{W}$. In the next section, we partly formalize these ideas.

# 4   RANK RECOVERY IN FINITE DEPTH NETWORKS

In this section, we study how the (approximate) rank of minimizer functions $f_{\hat{\mathbf{W}}}$ (i.e. functions at a global minimum $\hat{\mathbf{W}}$) for the MSE $\mathcal{L}_\lambda(\mathbf{W}) = \frac{1}{N}\sum_{i=1}^N (f_\mathbf{W}(x_i) - y_i)^2 + \frac{\lambda}{L}\|\mathbf{W}\|^2$ with data sampled from a distribution with support $\Omega$ is affected by the depth $L$. In particular, when the outputs are generated from a true function $f^*$ (i.e. $y_i = f^*(x_i)$) with $k = \mathrm{Rank}_{BN}(f^*;\Omega)$, we study in which condition the 'true rank' $k$ is recovered.

APPROXIMATE RANK 1 REGIME

One can build a function with BN-rank 1 that fits any training data (for example by first projecting the input to a line with no overlap and then mapping the points from the line to the outputs with a piecewise linear function). This implies the following bound:

**Proposition 2.** *There is a constant $C_N$ (which depends on the training data only) such that for any large enough $L$, at any global minimum $\hat{\mathbf{W}}$ of the loss $\mathcal{L}_\lambda$ the represented function $f_{\hat{\mathbf{W}}}$ satisfies*

$$\frac{1}{L} R(f_{\hat{\mathbf{W}}}; \sigma_a, \Omega, L) \leq 1 + \frac{C_N}{L}.$$

*Proof.* We use the same construction as in the proof of Theorem 1 for any fitting rank 1 function. $\quad\square$

This bound implies that the function $f_{\hat{\mathbf{W}}}$ represented by the network at a global minimum is approximately rank 1 both w.r.t. to the Jacobian and Bottleneck ranks, showing the bias towards low-rank functions even for finite (but possibly very large) depths.

**Jacobian Rank:** For any function $f$, the rescaled norm representation cost $\frac{1}{L}R(f;\Omega,\sigma_a,L)$ bounds the $L_p$-Schatten norm of the Jacobian (with $p=\frac{2}{L}$) at any point:

**Proposition 3.** *Let $f$ be a FPLF, then at any differentiable point $x$, we have*

$$\|Jf(x)\|_{2/L}^{2/L} := \sum_{k=1}^{\mathrm{Rank}Jf_{\mathbf{W}}(x)} s_k\left(Jf(x)\right)^{\frac{2}{L}} \leq \frac{1}{L}R(f;\Omega,\sigma_a,L),$$

*where $s_k\left(Jf_{\mathbf{W}}(x)\right)$ is the $k$-th singular value of the Jacobian $Jf_{\mathbf{W}}(x)$.*

Together with Proposition 2, this implies that the second singular value of the Jacobian of any minimizer function must be exponentially small $s_2\left(Jf_{\hat{\mathbf{W}}}(x)\right) \leq \left(\frac{1+\frac{C_N}{L}}{2}\right)^{\frac{L}{2}}$ in $L$.

**Bottleneck Rank:** We can further prove the existence of a bottleneck in the network in any minimizer network, i.e. a layer $\ell$ whose hidden representation is approximately rank 1:

**Proposition 4.** *For any global minimum $\hat{\mathbf{W}}$ of the $L_2$-regularized loss $\mathcal{L}_\lambda$ with $\lambda > 0$ and any set of $\tilde{N}$ datapoints $\tilde{X} \in \mathbb{R}^{d_{in}\times\tilde{N}}$ (which do not have to be the training set $X$) with non-constant outputs, there is a layer $\ell_0$ such that the first two singular values $s_1, s_2$ of the hidden representation $Z_{\ell_0} \in \mathbb{R}^{n_\ell\times N}$ (whose columns are the activations $\alpha_{\ell_0}(x_i)$ for all the inputs $x_i$ in $\tilde{X}$) satisfies $\frac{s_2}{s_1} = O(L^{-\frac{1}{4}})$.*

The fact that the global minima of the loss are approximately rank 1 not only in the Jacobian but also in the Bottleneck sense further supports our conjecture that the limiting representation cost equals the Bottleneck rank $R_\infty = \mathrm{Rank}_{BN}$. Furthermore, it shows that the global minimum of the $L_2$-regularized is biased towards low-rank functions for large depths, since it fits the data with (approximately) the smallest possible rank.

RANK RECOVERY FOR INTERMEDIATE DEPTHS

However, learning rank 1 functions is not always a good thing. Assume that we are trying to fit a 'true function' $f^* : \Omega \to \mathbb{R}^{d_{out}}$ with a certain rank $k = \mathrm{Rank}_{BN}\left(f^*;\Omega\right)$. If $k > 1$ the global minima of a large depth network will end up underestimating the true rank $k$.

In contrast, in the linear setting underestimating the true rank is almost never a problem: for example in matrix completion one always wants to find a minimal rank solution (Candès & Recht, 2009; Arora et al., 2019). The difference is due to the fact that rank 1 nonlinear functions can fit any finite training set, which is not the case in the linear case.

Thankfully, for large datasets it becomes more and more difficult to underestimate the rank, since for large $N$ fitting the data with a rank 1 function requires large derivatives, which in turn implies a large parameter norm:

**Theorem 2.** *Given a Jacobian-rank $k$ true function $f^* : \Omega \to \mathbb{R}^{d_{out}}$ on a bounded domain $\Omega$, then for all $\epsilon$ there is a constant $c_\epsilon$ such that for any BN-rank 1 function $\hat{f} : \Omega \to \mathbb{R}^{d_{out}}$ that fits $\hat{f}(x_i) = f^*(x_i)$ a dataset $x_1,\ldots,x_N$ sampled i.i.d. from a distribution $p$ with support $\Omega$, we have $\frac{1}{L}R(\hat{f};\Omega,\sigma_a,L) > c_\epsilon N^{\frac{2}{L}\left(1-\frac{1}{k}\right)}$ with prob. at least $1-\epsilon$.*

*Proof.* We show that there is a point $x \in \Omega$ with large derivative $\|Jf(x)\|_{op} \geq \frac{\mathrm{TSP}(y_1,\ldots,y_N)}{\mathrm{diam}(x_1,\ldots,x_N)}$ for the Traveling Salesman Problem $\mathrm{TSP}(y_1,\ldots,y_N)$, i.e. the length of the shortest path passing through every point $y_1,\ldots,y_m$, and the diameter $\mathrm{diam}(x_1,\ldots,x_N)$ of the points $x_1,\ldots,x_N$. This follows from the fact that the image of $\hat{f}$ is a line going through all $y_i$s, and if $i$ and $j$ are the first and last points visited, the image of segment $[x_i, x_j]$ is a line from $y_i$ to $y_j$ passing through all $y_k$s. The diameter is bounded by $\mathrm{diam}\Omega$ while the TSP scales as $N^{1-\frac{1}{k}}$ (Beardwood et al., 1959) since the $y_i$s are sampled from a $k$-dimensional distribution. The bound on the parameter norm then follows from Proposition 3. $\square$

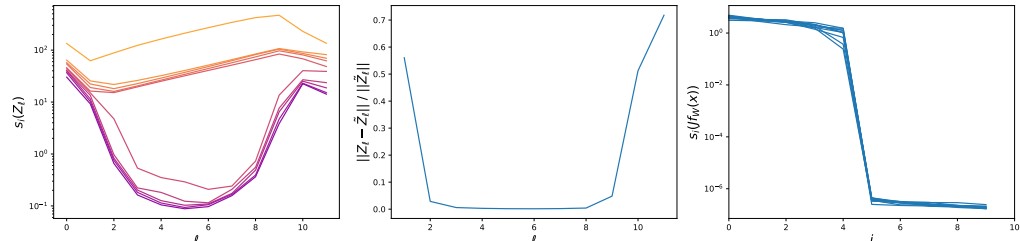

Figure 1: DNN (depth $L = 11$ and width $n_\ell = 300$) trained on a MSE task with rank 5 true function $f^* : \mathbb{R}^{50} \to \mathbb{R}^{50}$, with $N = 300$ and $\lambda = 0.05/L$. At the end of training, we obtain $\|W\|^2/L \approx 8$. **(left)** First 10 singular values of the matrix of activations $Z_\ell$ for all $\ell$. The representations are appr. rank 5 in the middle layers. **(middle)** The impact of the nonlinearity at each layer $\ell$, measured by the ratio $\|\tilde{Z}_\ell - Z_\ell\|_F / \|\tilde{Z}_\ell\|_F$ where $\tilde{Z}_\ell$ is the matrix of preactivations with entries $\tilde{\alpha}_k^{(\ell)}(x_i)$. This impact vanishes in the middle layers, supporting our intuition that the middle layers represent approximate identities. **(right)** First 10 singular values of the Jacobian $Jf_{\mathbf{W}}(x)$ at 10 random points.

This implies that the constant $C_N$ in Proposition 2 explodes as the number of datapoints $N$ increases, i.e. as $N$ increases, larger and larger depths are required for the bound in Proposition 2 to be meaningful. In that case, a better upper bound on the norm of the parameters can be obtained, which implies that the functions $f_{\hat{\mathbf{W}}}$ at global minima are approximately rank $k$ or less (at least in the Jacobian sense, according to Proposition 3):

**Proposition 5.** *Let the 'true function' $f^* : \Omega \to \mathbb{R}^{d_{out}}$ be piecewise linear with $\mathrm{Rank}_{BN}(f^*) = k$, then there is a constant $C$ which depends on $f^*$ only such that any minimizer function $f_{\hat{\mathbf{W}}}$ satisfies*

$$\frac{1}{L}R(f_{\hat{\mathbf{W}}}; \sigma_a, \Omega, L) \leq \frac{1}{L}R(f^*; \sigma_a, \Omega, L) \leq k + \frac{C}{L}.$$

Theorem 2 and Proposition 5 imply that if the number of datapoints $N$ is sufficiently large ($N > \left(\frac{k+\frac{C}{L}}{c}\right)^{\frac{kL}{2k-2}}$), there are parameters $\mathbf{W}^*$ that fit the true function $f^*$ with a smaller parameter norm than any choice of parameters $\mathbf{W}$ that fit the data with a rank 1 function. In that case, the global minima will not be rank 1 and might instead recover the true rank $k$.

Another interpretation is that since the constant $C$ does not depend on the number of training points $N$ (in contrast to $C_N$), there is a range of depths (which grows as $N \to \infty$) where the upper bound of Proposition 5 is below that of Proposition 2. We expect rank recovery to happen roughly in this range of depths: too small depths can lead to an overestimation of the rank[1], while too large depths can lead to an underestimation.

*Remark 5.* Note that in our experiments, we were not able to observe gradient descent converging to a solution that underestimates the true rank, even for very deep networks. This is probably due to gradient descent converging to one of the many local minima in the loss surface of very deep $L_2$-regularized DNNs. Some recent theoretical results offer a possible explanation for why gradient descent naturally avoids rank 1 solutions: the proof of Proposition 2 shows that rank 1 fitting functions have exploding gradient as $N \to \infty$, and such high gradient functions are known (at the moment only for shallow networks with 1D inputs) to correspond to narrow minima (Mulayoff et al., 2021).

Some of our results can be applied to local minima $\hat{\mathbf{W}}$ with a small norm: Proposition 3 implies that the Jacobian rank of $f_{\hat{\mathbf{W}}}$ is approximately bounded by $\|\hat{\mathbf{w}}\|^2/L$. Proposition 4 also applies to local minima, but only if $\|\hat{\mathbf{w}}\|^2/L \leq 1 + C/L$ for some constant $C$, though it could be generalized.

DISCUSSION

We now propose a tentative explanation for the phenomenon observed in this section. In contrast to the rest of the paper, this discussion is informal.

---

[1] Note that traditional regression models, such as Kernel Ridge Regression (KRR) typically overestimate the true rank, as described in Appendix D.1.

Ideally, we want to learn functions $f$ which can be factorized as a composition $h \circ g$ so that not only the inner dimension is small but the two functions $g, h$ are not 'too complex'. These two objectives are often contradictory and one needs to find a trade-off between the two. Instead of optimizing the bottleneck rank, one might want to optimize with a regularization term of the form

$$\min_{f = h \circ g} k + \gamma \left( C(g) + C(h) \right), \tag{1}$$

optimizing over all possible factorization $f = h \circ g$ of $f$ with inner dimension $k$, where $C(g)$ and $C(h)$ are measures of the complexity of $g$ and $h$ resp. The parameter $\gamma \geq 0$ allows us to tune the balance between the minimization of the inner dimension and the complexity of $g$ and $h$, recovering the Bottleneck rank when $\gamma = 0$. For small $\gamma$ the minimizer is always rank 1 (since it is always possible to fit a finite dataset with a rank 1 function in the absence of restriction on the complexity on $g$ and $h$), but with the right choice of $\gamma$ one can recover the true rank.

Some aspects of the proofs techniques we used in this paper suggest that large-depth DNNs are optimizing such a cost (or an approximation thereof). Consider a deep network that fits with minimal parameter norm a function $f$; if we add more layers to the network it is natural to assume that the new optimal representation of $f$ will be almost the same as that of the shallower network with some added (approximate) identity layers. The interesting question is where are those identity layers added? The cost of adding an identity layer at a layer $\ell$ equals the dimension $d_\ell$ of the hidden representation of the inputs at $\ell$. It is therefore optimal to add identity layers where the hidden representations have minimal dimension.

This suggests that for large depths the optimal representation of a function $f$ approximately takes the form of $L_g$ layers representing $g$, then $L - L_g - L_h$ identity layers, and finally $L_h$ layers representing $h$, for some factorization $f = h \circ g$ with inner dimension $k$. We observe in Figure 1 such a three-part representation structure in an MSE task with a low-rank true function. The rescaled parameter norm would then take the form

$$\frac{1}{L} \|\mathbf{W}\|^2 = \frac{L - L_g - L_h}{L} k + \frac{1}{L} \left( \|\mathbf{W}_g\|^2 + \|\mathbf{W}_h\|^2 \right),$$

where $\mathbf{W}_g$ and $\mathbf{W}_h$ are the parameters of the first and last part of the network. For large depths, we can make the approximation $\frac{L - L_g - L_h}{L} \approx 1$ to recover the same structure as Equation 1, with $\gamma = 1/L$, $C(g) = \|\mathbf{W}\|_g^2$ and $C(h) = \|\mathbf{W}_h\|^2$. This intuition offers a possible explanation for rank recovery in DNNs, though we are not yet able to prove it rigorously.

## 5    PRACTICAL IMPLICATIONS

In this section, we describe the impact of rank minimization on two practical tasks: multiclass classification and autoencoders.

### MULTICLASS CLASSIFICATION

Consider a function $f_{\mathbf{W}^*} : \mathbb{R}^{d_{in}} \to \mathbb{R}^m$ which solves a classification task with $m$ classes, i.e. for all training points $x_i$ with class $y_i \in \{1, \ldots, m\}$ the $y_i$-th entry of the vector $f_{\mathbf{W}^*}$ is strictly larger than all other entries. The Bottleneck rank $k = \mathrm{Rank}_{BN}(f_{\mathbf{W}^*})$ of $f_{\mathbf{W}^*}$ has an impact on the topology of the resulting partition of the input space $\Omega$ into classes, leading to topological properties typical of a partition on a $k$-dimensional space rather than those of a partition on a $d_{in}$-dimensional space.

When $k = 1$, the partition will be topologically equivalent to a classification on a line, which implies the absence of tripoints, i.e. points at the boundary of 3 (or more) classes. Indeed any boundary point $x \in \Omega$ will be mapped to a boundary point $z = g(x)$ by the first function $g : \Omega \to \mathbb{R}$ in the factorization of $f_{\mathbf{W}^*}$; since $z$ has at most two neighboring classes, then so does $x$.

This property is illustrated in Figure 2: for a classification task on four classes on the plane, we observe that the partitions obtained by shallow networks ($L = 2$) leads to tripoints which are absent in deeper networks ($L = 9$). Notice also that the presence or absence of $L_2$-regularization has little effect on the final shape, which is in line with the observation that the cross-entropy loss leads to an implicit $L_2$-regularization (Soudry et al., 2018; Gunasekar et al., 2018a; Chizat & Bach, 2020), reducing the necessity of an explicit $L_2$-regularization.

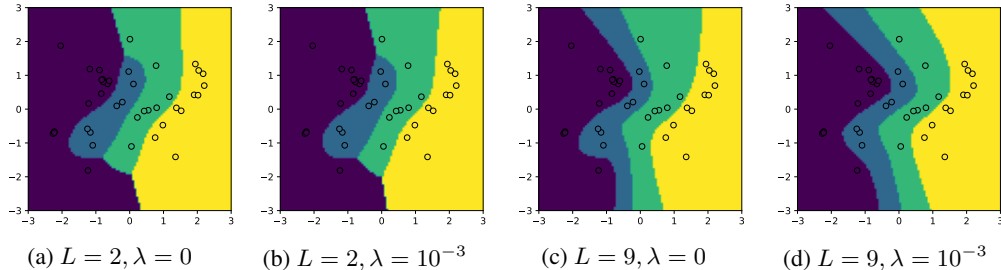

(a) $L = 2, \lambda = 0$    (b) $L = 2, \lambda = 10^{-3}$    (c) $L = 9, \lambda = 0$    (d) $L = 9, \lambda = 10^{-3}$

Figure 2: Classification on 4 classes (whose sampling distribution are 4 identical inverted 'S' shapes translated along the $x$-axis) for two depths and with or without $L_2$-regularization. The class boundaries in shallow networks **(A,B)** feature tripoints, which are not observed in deeper networks **(C,D)**.

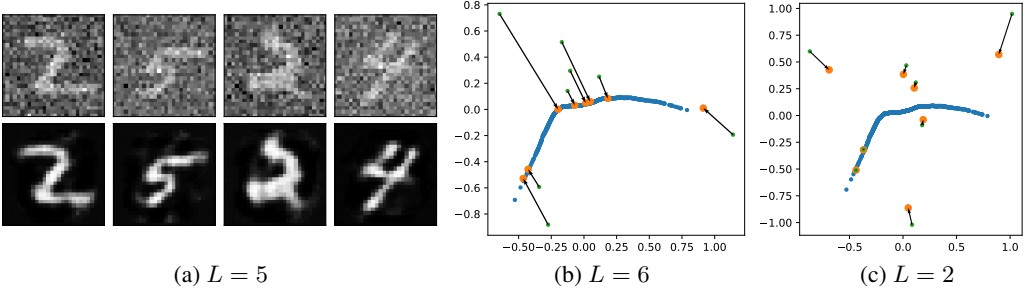

(a) $L = 5$    (b) $L = 6$    (c) $L = 2$

Figure 3: Autoencoders trained on MNIST **(A)** and a 1D dataset on the plane **(B, C)** with a ridge $\lambda = 10^{-4}$. Plot **(A)** shows noisy inputs in the first line with corresponding outputs below. In plots **(B)** and **(C)** the blue dots are the training data, and the green dots are random inputs that are mapped to the orange dots pointed by the arrows. We see that for large depths **(A, B)** the learned autoencoder is naturally denoising, projecting points to the data distribution, which is not the case for shallow networks **(C)**.

### AUTOENCODERS

Consider learning an autoender on data of the form $x = g(z)$ where $z$ is sampled (with full dimensional support) in a latent space $\mathbb{R}^k$ and $g : \mathbb{R}^k \to \mathbb{R}^d$ is an injective FPLF. In this setting, the true rank is the intrinsic dimension $k$ of the data, since the minimal rank function that equals the identity on the data distribution has rank $k$.

Assume that the learned autoencoder $\hat{f} : \mathbb{R}^k \to \mathbb{R}^k$ fits the data $f(x) = x$ for all $x = g(z)$ and recovers the rank $\text{Rank}_{BN}\hat{f} = k$. At any datapoint $x_0 = g(z_0)$ such that $g$ is differentiale at $z_0$, the data support $g(\mathbb{R}^k)$ is locally a $k$-dimensional affine subspace $T = x_0 + \text{Im}Jg(z_0)$. In the linear region of $\hat{f}$ that contains $x_0$, $\hat{f}$ is an affine projection to $T$ since it equals the identity when restricted to $T$ and its Jacobian is rank $k$. This proves that rank recovering autoencoders are naturally (locally) denoising.

## 6 CONCLUSION

We have shown that in infinitely deep networks, $L_2$-regularization leads to a bias towards low-rank functions, for some notion of rank on FPLFs. We have then shown a set of results that suggest that this low-rank bias extends to large but finite depths. With the right depths, this leads to 'rank recovery', where the learned function has approximately the same rank as the 'true function'. We proposed a tentative explanation for this rank recovery: for finite but large widths, the network is biased towards function $f$ which can be factorized $f = h \circ g$ with both a small inner dimension $k$ and small complexity of $g$ and $h$. Finally, we have shown how rank recovery affects the topology of the class boundaries in a classification task and leads to natural denoising abilities in autoencoders.

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
