# OpenReview forum: "Implicit Bias of Large Depth Networks: a Notion of Rank for Nonlinear Functions"
_ICLR.cc/2023/Conference — ICLR 2023 notable top 25%_

### Official Review · Reviewer_jUL5 · 2022-10-28

**Confidence:** 4
**Clarity, Quality, Novelty And Reproducibility:** The paper is well-written, clear, and…
**Correctness:** 4
**Technical Novelty And Significance:** 4
**Empirical Novelty And Significance:** 4
**Recommendation:** 10

**Strength And Weaknesses:**

Strengths:
This is a refreshing paper, which provides a novel and elegant perspective on the representation cost of functions by nonlinear, deep neural networks. I found the practical implications of the result particularly interesting -- specifically that deeper neural networks should find classification boundaries on a finite number of data points that do not have tripoints.

I wonder whether a finer quantitative statement can be made for multiclass classification of a finite number of data points N and finite depth L, in the style of Theorem 2 and Proposition 5 -- which bounds the number of tripoints in the minimum-representation-cost classification boundary.

There is also the intriguing open question about whether the upper bound is tight. And you can also ask about what happens if you add residual connections, which would seemingly lead to quite different behavior.

Weaknesses:
* I found the practical implications section relating to autoencoders a bit sparse, and would appreciate if more information could be added to clarify what is meant.
* In the proof of Theorem 1 in the appendix, maybe add a note that you are shifting f and g so that you only have to represent the identity on the upper quadrant (because Omega is bounded.) This is only mentioned in the proof of the second part of Theorem 1, and was a point of confusion for me before I read that.

**Summary Of The Paper:**

This paper studies the representation cost of piecewise linear functions by deep homogeneous nonlinear networks.

The representation cost of f is defined as R(f) = \min_{W : f_W = f} ||W||^2, where ||.|| is the L2 norm of the parameters and f_W is the neural network parametrized by W. This representation cost arises in the analysis of neural networks in a variety of settings, including training with cross-entropy loss, training with ridge regularization, and training with a small initialization.

1. The paper proves that in the limit of infinite depth L \to \infty, the representation cost of f : R^{d_{in}} \to R^{d_{out}} is sandwiched between two bounds: a lower bound based on the maximum rank of the Jacobian of the function at a point, and an upper bound called the "bottleneck" upper bound, based on the minimum inner dimension k such that one can write f = g \circ h, where g : R^{k} \to R^{d_{out}} and h : R^{d_{in}} \to R^{k}. Furthermore, \lim_{L \to\infty} R(f) / L satisfies several properties that one would like to have for a rank.

2. The paper also studies why, for large but finite depth L, the representation cost R(f) / L of the function f restricted to the training dataset does not trivialize to be approximately equal to 1 (Theorem 2).

3. Finally, practical implications are discussed, including qualitative differences between the classification boundaries learned by deep networks vs. shallow networks.

**Summary Of The Review:**

This was a very enjoyable and informative paper to read. It is novel, well-written, and interesting.

---

> ### Author Response · Authors · 2022-11-07
> **Author's response**
>
> Thanks for the review.
>
> We are not sure what you mean regarding finite data/depth results on the number of tripoints. Do you mean that there could be a regime where the number of tripoints is not exactly zero but only small in some sense? That's an interesting question. Assuming that for finite depths the BN rank $k$ of the learned function is larger than 1 but not too large, it might be the case that the number of tripoints would not be too large, since low dimensional classification boundaries probably have typically less tripoints than higher dimensional ones. Note however that in high dimension, there might be manifolds of tripoint, in which case it might not be obvious how to count their number.
>
> Along the same line, we are still unsure what would be characteristic of 2-dimensional (or k-dimensional) classification boundaries (in the same way that the absence of tripoints is characteristic of one dimensional boundaries). This would be something interesting to study.
>
> With Proposition 4, we are pretty close to proving that the upper bound is tight for BN-rank 1 functions, i.e. if $f$ has BN-rank 1 then $R_\infty(f)=1$, since we are able to show that there exists a bottleneck: a hidden representation that is approximately dimension 1. We think that to prove the general case, we will first need a generalization of Proposition 4 to the case $k>1$.
>
> You are right that residual connections lead to a completely different behavior: since the cost of representing the identity is always 0, regardless of the dimension, there is no pressure to minimize the inner dimension.
>
> We have clarified the section on autoencoders. We have also clarified the shifting of the outputs of $h$ in the proof of Theorem 1.

---

### Official Review · Reviewer_YMQ3 · 2022-10-29

**Confidence:** 3
**Correctness:** 4
**Technical Novelty And Significance:** 4
**Empirical Novelty And Significance:** 4
**Recommendation:** 8

**Clarity, Quality, Novelty And Reproducibility:**

The paper appears to be very novel and to open the door to an interesting direction. The paper is quite well written and generally clear.

There are a few things which make the reading of the proofs more difficult though, but that remains minor.

Details:

Some results and theorems do not repeat enough of the general concept to make them easy to read. For instance, proposition 2 doesn't explicitly mention what optimization problem is being learnt anymore. This forces the reader to read the paper quite linearly, which not everyone wants to do (a lot of the time we might be going back and forth between the supplementary and the main).

Figure 2 appears very early compared to the place where it is references, and the concept of "tripoint" is also only introduced later.

In theorem 2, the proof of point 4 is missing and the "point 4" in the proof is actually point 5.

In the beginning of the proof of Proposition 3 (page 3 of sup), a pointer to corollary 1 would be strongly appreciated. This applies to other places where this result is mentioned. I know that at least in one place it is clearly stated that this is a result of Arora et al., but in other parts of the paper it seems to be presented as if it was something the reader should consider obvious.

The proof of proposition 4 is a little hard. How do the authors get the first equation (especially the term $-\delta$)?

The statement at the end of the first paragraph of Section 4 (page 6 of sup) is not proved.

The proof of proposition is elegant, but it is hard to parse at first reading. There is a Lagrangian argument which is completely implied. The sums should be expressed less ambiguously (so one does not assume that the norms of the weights are also summed over layers)


Some places(e.g. Prop 7)  use $\|\|$  for the Froebenius norm, whilst others use $\|\|_F$, cf. Prop 4.

Section D1 could also be extended. Note that the explanations are for KR (Kernel Regression) rather than KRR as claimed there and in the main paper.





===============================minor typos============


In the (-1)th line of Proposition 4, $n_{\ell}$ should be replaced by $\ell_0$.

At the end of the second paragraph of section "Rank Recovery for Intermediate Depths", "rank one function can fit" should be "rank one functions can fit"

In page 2 of the supplementary, "finaly" should be "finally"

At the beginning of the proof of proposition 3, "there is a depth ..... networks which representing g..." should be "there is a depth... network which represents g"

At the third line of proposition 4 on page 3 of the sup, I think the $X_\ell$ (in the brackets) should be $X_{\ell}$.

At the beginning of the "upper bound" part of the proof of proposition 4, I think $\delta_0$ should be $\ell_0$.

In the middle of page 5 (still in the proof of Proposition 4), "equation 2" should be "equation (2)"

Page 6 of the sup has quite a few typos. Proposition 5, two lines after the main equation, the sentence should be split and doesn't make gammatical sense. For the last line, removing the "at" in "at any rank 1..." would make the sentence more coherent. There is also a space missing before the final inline equation. Same at the bottom of page 6 "and the network hat the end" should be "and the network h at the end". I would also usually prefer to put "resp.   " statements in brackets.

Prop 7: "...for some $\lambda>0$ then $W$ satisfy..."  ==> "...for some $\lambda>0$. Then $W$ satisfies..."

**Strength And Weaknesses:**

Strengths:

This is an extremely interesting topic at the forefront of AI theoretical research

The results themselves are very interesting and appear to break new ground (I am not familiar enough with the recent literature to  fully vouch for this though).

The paper is quite well written in general


Weaknesses:

Not much, but if one is being picky:

Some of the proofs should have more justifications

There are a few typos




**Summary Of The Paper:**

This paper proposes a notion of rank for non-linear functions, which is defined as the minimum possible $L^2$ norms of the weights of a neural network which matches the function, averaged over the layers, and asymptotically where the number of layers tends to infinity.
It is conjectured that this notion of rank is equivalent to that of "bottleneck rank", which is the minimum embedding dimension of an encoder decoder network that represents the function. The idea is that as the number of layers tends to infinity (and is much larger than the minimum required depth to represent the function), most layers are identity functions, which have Frobenius norm k where k is the embedding dimension at those layers. Hence minimizing the notion of rank defined in this paper becomes similar to minimizing the bottleneck rank.

It is shown that the rank defined here enjoys several sanity check properties with respect to compositions of functions etc. The first main theorem (Theorem 1) states that the rank defined here is sandwiched between the maximum rank of the jacobian of the function and the bottleneck rank. Later in Section 4, the paper studies the slightly more concrete situation of finite architectures. Proposition 2 is a non asymptotic version of Theorem 1, but it then leads onto proposition 4, which shows that if the ground truth rank is 1, then the global minimum of the regularized objective (corresponding to training a regularized neural network) has a very small ratio between the first two eigenvalues of its weights at at least one layer.

In the next section, a concern is raised: BN rank 1 functions are universal approximators of piecewise linear functions with a one dimensional output, thus the bias for low rank or rank one models could prevent one from learning the true rank. However, it is shown that for iid data, the cost of fitting the training set with a rank one function is prohibitively large for a large number of samples. It is therefore suggested that the depth should be chosen at an appropriate regime depending on the amount of data.

In the experiments, it is shown that deeper networks do learn "low-rank" functions at least in some cases (cf. figure 2). It is also shown that training deep neural networks on data of low BN rank results in a solution which contains many intermediate representations of rank approximately equal to the ground truth rank

**Summary Of The Review:**

To the best of my knowledge (I may not know all the relevant literature, which makes vouching for originality harder), this appears to be an excellent paper making non trivial advances in the fundamental understanding of the behavior of deep neural networks. I couldn't find any mistake in the proofs and read most of them (I had some trouble with proposition 4 though). The topic is also fascinating and the tone just right (explaining some intuition as well as proving some more rigorous theorems). I like that one can see the thought process of the writers as they performed the research. Some of the proofs should be polished and made longer with more thorough explanations.


I would give a score of 9 if the option was available.

---

> ### Author Response · Authors · 2022-11-07
> **Author's response**
>
> Thanks for the very thorough review!
>
> Thanks for all of your detailed remarks:
> - We have added more details in the statement of Proposition 2.
> - We have moved Figure 2 to the last page.
> - We have fixed the proof of Theorem 2.
> - We have added references to Corollary 1 when needed.
> - For the proof of Proposition 4, we have added a description of the general strategy/intuition and have better explained the first inequality.
> - By the end of section 4, do you mean the KRR rank question? We have improved the discussion of the Jacobian rank of KRR.
> - We have clarified the proof of Proposition 7.
>
> We have also fixed all of the typos you mentioned.

---

### Official Review · Reviewer_44KN · 2022-11-01

**Confidence:** 3
**Correctness:** 3
**Technical Novelty And Significance:** 2
**Empirical Novelty And Significance:** 2
**Recommendation:** 5

**Clarity, Quality, Novelty And Reproducibility:**

## Clarity and quality
The paper is not very well written and at some points hard to follow. The authors have assumed the reader is knowledgeable with works in this area and the proofs are very terse making them hard to understand.

## Novelty
The results are novel to the best of my knowledge. However, I believe the authors should make a better case why these results are interesting and useful.

## Comments
 - Proof of Proposition 2 is quite long. I suggest adding a proof overview either in the appendix or the main body.  Also, $L$ clearly needs to be larger than something for this theorem to hold which needs to be added to the statement of the theorem for correctness.

 - Why is $\epsilon$ not showing up in the bound of Theorem 2? Is there a typo?

 - In the appendix B, in proof of Prop. 2 (Prop. 3 of the main) the authors use a result from Soudry et al. which I could not find in the paper that is cited. Please add a note which Theorem or Proposition of this paper implies this result.

## Minor comments and typos
 - Page 1, 2nd paragraph from the bottom: in $f(x) = w^\top x$, $x$ is missing. There's also a typo in cost for the fully connected networks.
 - Page 2, definition of Jacobian rank should use $\max$ instead of $\min$
 - Page 4, under the properties of rank: "Property 2 implies that Rank is invariant under pre-and post-composition with bijections..." It is not clear to me how property 2 implies that. It needs more explanation.

**Strength And Weaknesses:**

## Strengths
 - The paper looks at the implicit bias that depth of fully connected networks induce. This problem is interesting and different implicit biases have attracted a lot of attention in the past few years.

 - The paper tries to characterize this implicit bias theoretically and shows a few examples of what these theoretical results might imply in practice.

## Weakness
 - The paper is hard to follow. At times it is not clear how one results implies the other. The proofs in the main paper are very terse and even following some of the proofs in the appendix are not easy.

**Summary Of The Paper:**

The paper introduces two notions of rank for nonlinear functions, the Jacobian rank, and the bottleneck rank. The authors then consider fully connected neural networks with homogeneous nonlinearities. They first show that for $L\rightarrow\infty$, the reconstruction cost of any piecewise linear function is sandwiched between the two notions of rank. Next, they show several results regarding the reconstruction cost of any minimizer of the $\ell_2$-regularized empirical risk minimization problem when $L$ is large but finite. They have a discussion on some of the implications of these theoretical results.

**Summary Of The Review:**

The paper looks at the problem of implicit bias of fully connected neural networks in the large depth regime and introduces notions of rank that can explain the nature of the implicit bias for deep enough networks. This problem might be interesting and worthwhile to look at, however the results are not well presented and unless the conjecture the authors present is correct, the results do not directly imply there is a bias towards smaller bottleneck ranks.

---

> ### Author Response · Authors · 2022-11-07
> **Author's response**
>
> Thanks for your review. We were able to address some of your complaints, but we are unsure about some of them:
> - What are the previous works that needed to be better explained?
> - The proof of Proposition 2 is not very long, do you mean Proposition 4? We have added a paragraph describing the general strategy for the proof of Proposition 4. You are right that for Proposition 2, $L$ needs to be large enough, we have added that.
> - In Theorem 2, $\epsilon$ appears in the constant $c$ (which have now rewritten $c_\epsilon$ to emphasize this dependence).
> - Sorry this was a mistake on our part, we cited the wrong paper, we have now updated the citation.
>
> Thanks for pointing to those typos, we have corrected them, we were however not able to identify where the typo with the definition of the fully-connected cost lies.
>
> Regarding the invariance under post- and pre-composition with bijections, we apologies for this mistake, this actually follows from Property 2,3 and 5 instead of only 2. Note however that in the newer version, we have removed Property 2 since it follows from 3 and 5. There are now only 4 properties and Property 3 has become Property 2.
> The proof goes as follows: for any bijection $\phi$ on $\mathbb{R}^{d_{in}}$, we have $d_{in} = \mathrm{Rank}(\phi \circ \phi^{-1}) \leq \min [\mathrm{Rank}\phi, \mathrm{Rank}\phi^{-1}]\leq d_{in}$, thus $\mathrm{Rank}\phi = \mathrm{Rank}\phi^{-1} = d_{in}$. As a result $\mathrm{Rank}(f \circ \phi)\leq \min [\mathrm{Rank}f, d_{in}]=\mathrm{Rank}f$  and $\mathrm{Rank}f = \mathrm{Rank}(f\circ\phi\circ\phi^{-1})\leq \mathrm{Rank}(f\circ\phi)$. We will add a proof in the appendix.
>
> Finally we do not agree that without the conjecture, there is no proof of a bias towards low BN-rank solutions, we do show that the global minima of very deep networks have an approximate rank 1 hidden representation, which implies that they are approximately BN-rank 1.

---

### Official Review · Reviewer_JJ6D · 2022-11-04

**Confidence:** 4
**Clarity, Quality, Novelty And Reproducibility:** 1. Clarity
**Correctness:** 4
**Technical Novelty And Significance:** 4
**Empirical Novelty And Significance:** 3
**Recommendation:** 8

**Strength And Weaknesses:**

Strengths

1. The generalization of the notion of rank to regular but nonlinear function is indeed a very difficult task to attempt in general. I would characterize this paper as a sensible and thought-provoking attempt since the resulting notion is not trivial in certain regimes (large dataset, large depth, etc.).
2. Traditional low rank literature usually requires linear separability or low rankness of the data matrix or whitened input. This paper does not make such assumptions.
3. The theoretical analysis is done rigorously, with great intuition and explanation (although the Discussion subsection in section 4 is informal, it helps getting the general idea of the proof and feasible future directions).
4. Experiments and applications well-support the theoretical findings.

Weaknesses
1. The rank characterization in the main results are done for some global minimizer of the regularized ERM, and not what is actually learnt by specific algorithms such as gradient descent or stochastic gradient descent. In practice, it is very hard for GD/SGD to find said global minimizer.
2. The infinite depth regime is not something one expects to see in practice.
3. Theorem 2 only addresses that the representation cost for rank-1 functions would be high if the true function has rank k. It is not clear how the representation cost penalty interplays with the underestimation of rank (for example, can we get some kind of high cost guarantee for rank-m functions where 1 < m < k?) The point made in the section would be a lot more convincing if this interplay is made quantitative.

Additional remarks and questions

1. The definition of Jacobian rank in the paper can be problematic, especially with the use of ReLU/piecewise linear function since the Jacobian is not defined everywhere. Changing this to maximum rank over all affine pieces should address this problem (but would limit future applicability to neural networks that are not piecewise linear). The most general way to address this would probably involve defining the Jacobian rank for an appropriate subdifferential model (say Clarke), but this requires a lot more work.
2. Theorem 1 has a reference to Proposition 3, which only appears later in the paper.
3. In page 4, section 3, the author claims that “This result (of Theorem 1) suggests that large-depth neural networks are biased towards functions which have a low Jacobian or bottleneck rank…”. However, assuming that the representation cost of the optimal function is low (which is reasonable given that it is a term in the optimization objective), only the Jacobian rank is constrained to be small. The bottleneck rank can be arbitrarily large (unless the author’s conjecture that the representation cost in the infinite depth limit equals the bottleneck rank is true).
4. The use of TSP in the proof of Theorem 2 comes as quite a surprise and it would be more intuitive if there are some discussion of this proof technique (also, it’s not immediately clear what metric space the TSP is being formulated in). What conditions are required for the scaling of Beardwood et al 1959 to hold? (If all the y_i’s are in a straight line, I don’t suppose this scaling holds?)

**Summary Of The Paper:**

This paper introduces two new notions of rank for nonlinear functions (Jacobian rank and bottleneck rank). These definitions satisfy a set of properties of matrix ranks and thus generalize this classical notion. Moreover, under these rank notions, there exists regimes (large depth, large sample size) where the authors showed that neural networks that minimize a regularized ERM objective have low rank (and sometimes may recover the rank of the true teacher function) - suggesting a form of implicit bias. Experiments are done to support the claim and application to autoencoders are discussed.












**Summary Of The Review:**

The paper has very interesting and strong theoretical results; and potentially opens doors to a lot more directions in understanding how deep learning works rigorously. I therefore recommend acceptance in anticipation that the writing is touched up a bit more in the final version, as well as more discussions on techniques/results.

---

> ### Author Response · Authors · 2022-11-07
> **Author's response**
>
> Thanks for the thoughtful review.
>
> We generally agree with the weaknesses that you mention, but we strongly believe that they can be overcome, and are working on solving them:
> 1) We believe that this type of bottleneck structure could extend to (at least some of the) local minima of the loss, possibly with different inner dimensions. At the very least those with a parameter norm close to that of the global minimum. Assuming that the learned network is close to a local minimum of the $L_2$-regularized loss is a much more reasonable assumption.
> 2) At the moment, we were able to prove approximate results for finite but large widths (which is already a finite width result), but we believe that for sufficiently large depths the  minimizer functions (those represented by global minima) have low BN rank (not approximately so). The minimal depth required to observe a Bottleneck structure probably depends heavily on the structure of the problem/function learned and it is difficult to characterize such structure. In our experiments, we observed a pretty good agreement with our theory already for depths around 10, though these were of course small datasets.
> 3) We think that a more general result applies: that to fit a rank $k$ function with a rank $m<k$ function on $N$ random datapoints, one would need a representation cost of order $N^{\frac{1}{L}(\frac{1}{m}-\frac{1}{k})}$, but this seems to require a completely different argument that the one based on the Traveling Salesman Problem and we were not able to prove it yet.
>
> This paper is only a first step, it focuses on the bias of the global minima of infinite depth and large depth networks, and already a lot of interesting behavior arises in this setting.
>
> Additional remarks:
> 1) We have changed the definition of the Jacobian rank to address your issues, one can take the maximum over all points $x$ where $f$ is differentiable. Note that this is equivalent to the Clark differential definition since the Clark Jacobian at a non-differentiable point $x$ equals the limit of the Jacobian at a sequence of differentiable points $x_1,\x_2,\dots$ and the rank of the limit is upper bounded by the limit of the rank.
> 2) We have added a reference to the section Proposition 3 appears in, we are too restricted in terms of page number to restate Proposition here.
> 3) You are right, we have added a mention that `assuming that our conjecture is true', a low norm implies a low BN-rank. Note that at least in the case of the global minima, we actually show that the (approximate) BN rank is low.
> 4) We have added a small explanation in the sketch of proof for the appearance of the TSP. Also, the metric is the Euclidean one. For the result to hold, we need the $y_i$s to be sampled i.i.d. from a $k$-dimensional distribution, which follows from the fact that $f^* $ has Jacobian rank $k$ and that the distribution of the $x_i$s on $\Omega$ is dense (note that in the original version of the paper, we assumed that the BN rank was $k$ which is insufficient for this proof, we have corrected it). The randomness of the $y_i$s ensures that they do not end up on a line with high probability.
>
> We will add a section in the Appendix to describe the setup of the numerical experiments in details for reproducibility.

---

### Official Review · Reviewer_BGRP · 2022-11-04

**Confidence:** 3
**Correctness:** 3
**Technical Novelty And Significance:** 3
**Empirical Novelty And Significance:** Not applicable
**Recommendation:** 6

**Clarity, Quality, Novelty And Reproducibility:**

There are some issues with clarity caused by imprecise references, typos (both
in language and in proofs), and missing details in mathematical arguments. The
problem considered is rather novel, and the authors provide technical arguments
in the appendices.



**Strength And Weaknesses:**

## Strengths

The work considers an important and challenging problem in the study of
implicit bias of interpolating neural networks -- the case of nonlinear neural
networks of depth larger than two -- which has not been successfully treated in
any significant generality in any of the prior works in this area.

## Weaknesses

The results are all phrased in terms of "representation costs of piecewise
linear functions for homogeneous neural networks of sufficient width", where
the width needs to depend on the target function $f$ -- in particular the
definition at the bottom of page 3 does not pertain to any fixed class of
architectures. This makes it hard to compare to prior work, e.g. most relevant
is by Ongie et al. (2020) -- why not just formulate everything in terms of
"infinite width" networks as Ongie et al. do? This would seem to greatly
facilitate comparison. The authors draw an analogy to the
rank minimization implicit bias in deep linear networks throughout the paper,
but it seems worth noting in this connection that the authors' results seem
much weaker (for deep linear networks, the representation cost characterization
holds for very general width-depth configurations, rather than only the case of
depth $\gg$ width here).

In the proofs, one sees that the limiting representation cost (imagine it in
terms of the bottleneck rank, since the lower bound in terms of the Jacobian
rank seems rather coarse -- it doesn't capture any information about the
complexity/variability/number/etc. of the different piecewise linear components
of $f$, only their maximum slopes), is realized by a network that is infinitely
deep, and has fixed width (although this width in general depends on $f$). Such
networks are impossible to learn with gradient descent -- e.g. one thinks of
the dying ReLU issue http://arxiv.org/abs/1903.06733. Could the authors comment
on why conclusions drawn from this limit can still be relevant for practical
networks? It seems that the theory (including in section 4, where "finite-depth
networks" are considered) does not have any implications for networks whose
width is growing together with the depth. Similarly, the upper bound developed
in the proofs on the representation cost seems to have no ability to capture
interesting practical phenomena in the study of representation by deep
homogeneous networks such as depth separations (where a certain function $f$
can be represented with far greater efficiency by a network of $L+1$ layers
than a network of $L$ layers).

There are some technical imprecisions that make it hard to assess the
correctness of the theory (see below for some more minor ones).
- Since piecewise linear functions are not generally differentiable, it would
  sense to comment quickly on this issue when defining the Jacobian rank (since
  a maximum is taken over $\Omega$, involving points where it will not be
  defined).
- Proof of Theorem 1 (second inequality): the argument seems to be using
  implicitly the claim "if a piecewise linear $f$ can be written as $f = g
  \circ h$, then $g$ and $h$ are also piecewise linear". This does not seem to
  be true -- consider any diffeomorphism $\phi$ on $h(\Omega)$, then $f = (g
  \circ \phi^{-1}) \circ (\phi \circ h)$ gives another decomposition of $f$
  into two functions that need not be piecewise linear. The claim seems more
  subtle than it is treated in the proof (it does not seem immediate to me that
  all such decompositions are related in this way).
- It is not made clear in the main body that Theorem 1 is only shown to satisfy
  the bijection property for piecewise linear bijections.


## Minor / Questions

Last line in first graf of "Contributions" on page 2: it would seem to be
preferrable to actually state the five properties here--if they are somewhat
technical, perhaps in "informal" form?

I was trying to find the references for the representation costs of linear
networks quoted in the introduction and in the appendices, but I could not find
anywhere in the cited references (the authors give Gunasekar 2018, Moroshko
2020, Soudry 2018) the specific results that were being invoked. It would be
better if an actual specific pointer to where in these papers the results being
invoked can be found were added, or a more direct reference was included (for
example,
https://proceedings.neurips.cc/paper/2021/hash/e22cb9d6bbb4c290a94e4fff4d68a831-Abstract.html
seems to have the requisite results).

Some technical imprecisions:
- Section 1 defines the Jacobian rank with a minimum, but Definition 1 defines it
  with a maximum.
- Given the context, it seems the authors are focusing on the class of
  piecewise linear functions under consideration those with finitely many
  pieces (otherwise it does not make sense to me to write $\max$ for the
  definition of the Jacobian rank rather than $\sup$).
- Page 4 after five properties: I cannot see how property 2 implies invariance
  under composition with bijections (property 2 does not involve a statement
  about composition). It is not clear to me how to deduce this from these five
  properties even thinking of linear maps and their rank function.


**Summary Of The Paper:**

the authors consider the implicit bias of deep neural networks with homogeneous
activations and linear layers, trained to minimize the square loss against a
given piecewise linear target function with $\ell^2$ regularization. previous
works have explored the functions achieving the minimum "representation cost"
(the minimum weight norm of a network that interpolates the data) in simpler
cases, such as linear networks and depth-two relu networks. the authors
consider a representation cost that is minimized over all networks of
sufficiently large widths, and show that in the asymptotics where the network
depth goes to infinity, the normalized representation cost converges to a
suitable nonlinear notion of "rank" that the authors define. they study various
perturbations off this limit (finite but large depth, on a loss taken over $n$
empirical samples), and interpret the resulting conclusions in positive and
negative lights of this "low-rank bias". simple experiments are presented that
connect to classification and autoencoding problems.



**Summary Of The Review:**

The work is very creative, and the connections between the bottleneck rank and
the training of neural networks for classification and autoencoding mentioned
in the last section of the paper is very interesting. However, the theoretical
techniques do not seem to have much relevance to the regimes encountered in the
training of practical networks, given that the theory seems to apply only to
networks of untrainable depths and the representation cost defined takes an
infimum over architectures (widths), which makes it challenging to conclusively
link the work's contributions to these applications.

---

> ### Author Response · Authors · 2022-11-07
> **Author's Response**
>
> Thanks for the thorough review and the many relevant questions/remarks.
>
> 1) Sufficient width: The sufficiently wide DNNs assumption can be thought as not only showing the infinite-width behavior but also proving a $O(0)$ rate of convergence to this limit, in the sense that there is a smallest width $w_0$ such that our results apply to any network which is wider than $w_0$, including of course infinitely wide networks. In the finite data case, we have an explicit upper bound on how wide the network needs to be $w_0 \leq N(N+1)$, in the infinite data case we can have an upper bound in terms of the number of linear regions not only of $f$ but also of its factorization $g$ and $h$ (for the upper bound in Theorem 1 for example), making it less explicit. When it comes to the width/depth interaction, in all of our results, we have a minimal $w_0$ (which does not depend on the depth) and a minimal depth $L_0$ which might depend on $w_0$ but not on the width $w$. All the constant that are independent of the widths $w$. Our results therefore apply to any limits $L,w \to \infty$ regardless of how fast they grow compared to one another. We will add a discussion in the appendix to clarify this and make the values of $w_0$ and $L_0$ explicit in the proofs.
>
> 2) Coarseness of the lower bound: We agree that this lower bound is coarse in general, however we believe that global minima (and possibly local minima too) of large depth networks have a bottleneck structure where there are a growing number of `almost' identity layers in the middle of the network. In these layers the datapoints are in the upper quadrant $\mathbb{R}^{n_\ell}_+$ where the nonlinearity $\sigma_a$ equals the identity and where the slope is maximal. In the presence of such a structure, the lower bound in terms of the Jacobian is tight for large depths. And Theorem 1 does show that when fitting a function of the form $\psi \circ A \circ \phi$, the lower bound is tight up to $O(1/L)$ terms.
>
> 3) Learnability of narrow networks: We agree that very deep and narrow networks are in general hard to train. However note that only the `effective width' of the network is small, not the actual width. As described in more details in (Jacot et al., 2022b) if a global minimum is attained at a certain width $(n_0,n_1,\dots,n_L)$ it can be attained at any larger width $n'_\ell \geq n_\ell$ (by essentially adding copies of already existing neurons in the hidden layers, or adding dead neurons). The idea is that one can train a very wide network (taking advantage of the large width to avoid convergence problems) and end up learning a network which is equivalent to a much narrower network (which should generalize better thanks to the smaller effective number of parameters). At the moment, there is no proof of this phenomenon in the deep case, but the mean-field limit (Bach 2017; Chizat and Bach, 2018; Chizat and Bach, 2020) suggests that this is the case in shallow networks, since the global minima are known to be sparse in some settings and convergence to these local minima can be guaranteed for infinitely wide networks.
>
> 4) The relation between our result and depth separation results is very interesting. Our intuition is that depth separation result should become more and more rare the deeper the network (functions that can be represented in a significant more efficient manner by a depth 101 in comparison to a depth 100 network are probably extremely complex and uncommon). We think of our result as describing the representation cost of DNNs past these separation results, once there is no significant advantage to increasing the depth (in this regime increasing the depth leads to adding more identity layers, instead of completely changing the representation). Of course understanding when exactly this regime starts is still a completely open question.

---

> > ### Comment · Reviewer_BGRP · 2022-12-06
> > **thanks**
> >
> > Thanks to the authors for the thorough rebuttal and for fixing issues of precision in the manuscript. I still harbor some reservations about the significance of this regime, but it is clear that the work connections to many very interesting research topics and will lead to interesting follow-up work, even with the somewhat challenging presentation in the submission now. I will therefore increase my score.

---

> ### Author Response · Authors · 2022-11-07
> **Smaller points:**
>
>
> - We have clarified in the main that for piecewise linear functions the maximum in the Jacobian rank definition can be taken over all linear regions (for any point in the inside of these regions).
> - We have specified that our results apply to finite piecewise linear function. This is indeed very important and we apologies for the unclarity. Note that even for general functions, the max is still justified since the rank of the Jacobian at any point $x$ takes values in the finite set $[0,1,\dots,\min(d_{in},d_{out})]$.
> - We apologies for this mistake, this actually follows from Property 2,3 and 5 instead of only 2. Note however that in the newer version, we have removed Property 2 since it follows from 3 and 5. There are now only 4 properties and Property 3 has become Property 2.
> The proof goes as follows: for any bijection $\phi$ on $\mathbb{R}^{d_{in}}$, we have $d_{in} = \mathrm{Rank}(\phi \circ \phi^{-1}) \leq \min [\mathrm{Rank}\phi, \mathrm{Rank}\phi^{-1}]\leq d_{in}$, thus $\mathrm{Rank}\phi = \mathrm{Rank}\phi^{-1} = d_{in}$. As a result $\mathrm{Rank}(f \circ \phi)\leq \min [\mathrm{Rank}f, d_{in}]=\mathrm{Rank}f$  and $\mathrm{Rank}f = \mathrm{Rank}(f\circ\phi\circ\phi^{-1})\leq \mathrm{Rank}(f\circ\phi)$. We will add a proof in the appendix.
>
> Minor/Questions:
>
> We now state the 4 properties in the contribution section. We did indeed put the wrong reference for the representation cost of linear fully-connected nets, we have fixed this. We have fixed the other issues you mentioned.

---

### Official Review · Reviewer_mVym · 2022-11-05

**Confidence:** 5
**Clarity, Quality, Novelty And Reproducibility:** Very well.
**Correctness:** 4
**Technical Novelty And Significance:** 4
**Empirical Novelty And Significance:** 4
**Recommendation:** 8

**Strength And Weaknesses:**

Strengths:
1. This paper is very well written; the notations are clear and very carefully designed; the organization of this paper is very clear.
2. The topic of rank in deep networks are very important for a broad range of domains. Many previous works are not as successful in establishing a general enough yet elegant theory analysis framework for this topic. The three metrics, representation cost, Jacobi rank, and BN-Rank in this work are convincing, general, and elegant to use.
3. Suffcicent numerical studies validates the theory justifications very well.
4. Theory results on the connections between ranks and representation costs are very interesting and compelling.


I have the following questions:

1. In the representation cost, the norm || || is spectral norm or what?
2. In the proof to Th1, do we assume that f tends to have a constant dimension for most of the intermediate layers? I think this makes sense for neural networks constructed manually, but may not be that reasonable for the underlying target function, the property of which is unknown.
3. In definition 1, I think perhaps a better intuition for this definition of rank would be, considering Sard's Theorem and the Rank theorem of manifolds, only those regions of the highest rank are of true influences in the output manifold, as in [1].
4. Some previou work[1] also discusses rank behavior of network ranks from the opinion of random matrix theory. The problem of using Jacobi rank directly is that, the rank of matrix is instable under small noises and errors, thus it is impossible to measure them in practice. So perhaps using the counting measure of significant singular values could be better.

[1] Rank Diminishing in Deep Neural Networks, NeurIPS2022.

**Summary Of The Paper:**

This paper studies the rank behavior of deep neural networks with three different types of rank definitions: the maximum of network Jacobi rank, bottlenet rank, and representation cost. The authors rigorously demonstrate the intrinsic connections among those three concepts, revealing a fascinating property of general deep learning models. The authors also show the representation cost of  BN rank-1 and rank-k functions. The results in this paper are appealing to a broad audience in machine learning.

**Summary Of The Review:**

See comments above.

---

> ### Author Response · Authors · 2022-11-07
> **Answer to questions**
>
> Thanks for your review and for the relevant link.
>
> Regarding your questions:
> 1) The parameters $\mathbf{W}$ are thought as a vector obtained by the concatenation of all the weight matrices and bias vectors. The norm in the representation cost definition is then simply the Euclidean norm. We therefore have $|| \mathbf{W} ||^2 = \sum || W_i ||_F^2+|| b_i ||^2$.
> 2) Though we generally believe that it might actually be the case that the global minima of large depth networks have many low-dimension almost identity layers in the middle, you are right that we do not prove this in this paper. For the proof of Theorem 1, we only show that there is a network with this structure that represent $f$ and whose rescaled norm converges to the rank $k$. In settings where the Jacobian and Bottleneck ranks match, we know that this constructed network is almost optimal, but this does not tell us anything about how the optimal network looks like. While Theorem 1 does not describe the structure of the optimal network, Proposition 4 does show that the optimal network must have at least one almost rank 1 layer.
> 3) Thanks for the references and link, this is indeed a very nice way to motivate the Jacobian rank. We have added it to the main.
> 4) You are right, and in addition to the numerical issue, there is also the issue that in practice the parameters are never exactly at a local/global minimum but very close to it, and the rank can change within this small neighborhood. Note that this is the reason why we plotted the first 10 eigenvalues in Figure 1 (a) and (b), there is a clear drop after the first 5 eigenvalues, thus suggesting an effective rank of 5. But we could also have counted the number of eigenvalues above a threshold.

---

### Decision · Program_Chairs · 2023-01-20

**Decision:**

Accept: notable-top-25%

**Justification For Why Not Higher Score:**

This can be an oral. The reason I only gave a spotlight was because there were some concerns about correctness (even though I feel the author response has addressed most of them, I haven't checked the entire proof).

**Justification For Why Not Lower Score:**

The results are very interesting I think it deserves at least a spotlight.

**Metareview: Summary, Strengths And Weaknesses:**

This paper considers several different definitions of rank for neural networks, including Jacobian rank, bottleneck rank and representation cost. The paper proved some interesting connections between these rank definitions, and considered the representation power of models with limited rank. Most reviewers find the results novel and very interesting. There are some minor concerns about clarify but they seem to be addressed in author response.

**Note From Pc:**

if the above contains the word "oral" or "spotlight" please see: "oral" presentation means -> notable-top-5% and "spotlight" means -> notable-top-25%. As stated in our emails, we are disassociating presentation type from AC recommendations